# Carbon Dots—Types, Obtaining and Application in Biotechnology and Food Technology

**DOI:** 10.3390/ijms241914984

**Published:** 2023-10-07

**Authors:** Joanna Szczepankowska, Gohar Khachatryan, Karen Khachatryan, Magdalena Krystyjan

**Affiliations:** 1Faculty of Biotechnology and Horticulture, University of Agriculture in Krakow, Al. Mickiewicza 21, 31-120 Krakow, Poland; j.szczepankowska17@wp.pl; 2Faculty of Food Technology, University of Agriculture in Krakow, Al. Mickiewicza 21, 31-120 Krakow, Poland; gohar.khachatryan@urk.edu.pl (G.K.); karen.khachatryan@urk.edu.pl (K.K.)

**Keywords:** carbon dots, graphene carbon dots, carbon quantum dots, carbon nanodots, carbonized polymer dots

## Abstract

Materials with a “nano” structure are increasingly used in medicine and biotechnology as drug delivery systems, bioimaging agents or biosensors in the monitoring of toxic substances, heavy metals and environmental variations. Furthermore, in the food industry, they have found applications as detectors of food adulteration, microbial contamination and even in packaging for monitoring product freshness. Carbon dots (CDs) as materials with broad as well as unprecedented possibilities could revolutionize the economy, if only their synthesis was based on low-cost natural sources. So far, a number of studies point to the positive possibilities of obtaining CDs from natural sources. This review describes the types of carbon dots and the most important methods of obtaining them. It also focuses on presenting the potential application of carbon dots in biotechnology and food technology.

## 1. Introduction

Nowadays, among the most exciting and fast-growing fields of science is nanotechnology, finding applications in various areas of life, for such areas as transportation, food, medicine, electronics or cosmetics [1,2]. It can be described as a group of techniques involving the control of objects at the nanometer scale to generate materials and devices with novel and unique properties [3]. The unusual properties of nanomaterials are due to their small size, below which quantum effects begin to significantly affect the behaviour of the molecule. Factors such as the size and shape of the particular structural elements can affect the nature of a particle more than its chemical composition. The higher surface-to-volume ratio and large specific surface area in nanometric materials indicate that their features are primarily determined by atoms and interactions located on their outer layer, instead of deep within the phase, as is with macroscopic objects. In addition, the number of chemical bonds is reduced, so that individual bonds gain a greater influence on the functioning of the nanostructure [4]. The production of new types of carbon-based nanomaterials, the exploration of their properties and their later functionalization are major tasks of nanotechnology and materials engineering. Carbon nanoparticles are most commonly used as nanofillers in composite materials, since this combination can strongly enhance their functionality [5,6,7]. Their essential properties include very good electrical and thermal conductivity, electrochemical and thermal stability, and mechanical strength [8]. Recent scientific reports have also demonstrated their high biocompatibility with cells such as osteocytes and neurons, as well as their antimicrobial activity, significantly increasing their use in biomedical science [9,10]. There are several allotropic carbon varieties. The best known are nanotubes, fullerenes, graphene and carbon dots. Because of differences in the spatial distribution of the elements, each variant has unique physical properties and chemical activity. Over the past few years, carbon quantum dots (CQDs) have gained particular interest among researchers as a new class of carbon nanomaterials. Low toxicity, a wide variety of both physical and chemical synthesis methods, and biocompatibility make carbon dots a promising replacement for semiconductor quantum dots [11,12,13,14].

A further advantage of carbon dots is the extremely simple and environmentally friendly way of obtaining them. They can be synthesized from almost all organic materials, including bio-waste generated in huge quantities. Using pollutants to produce high-functional materials is a solution for developing a sustainable, closed-loop economy [7,15]. Using different sources and modifying the parameters for synthesizing carbon dots can significantly affect their properties [16,17]. Similarly, the methods of obtaining and the ratios of the various components will have a considerable effect on the characteristics of the nanocomposite. Therefore, in order to find a material with the desired abilities, it is necessary not only to test the mutual influence of the individual components that build the composite, but also to select appropriate synthetic routes [18,19,20].

## 2. Carbon Dots

Carbon is an element that has many allotropic varieties. Each is characterized by different physicochemical and biological properties. In recent years, carbon dots (CDs) have attracted considerable interest. This term is used to describe a wide range of fluorescent carbon nanomaterials with sizes below 10 nm [21]. A rich source resource and numerous synthetic strategies ensure their great diversity. Features common to all forms of CD are lack of toxicity, well-defined mechanical properties, easy surface functionalization, stability at high temperatures, excellent electrical conductivity, barrier properties, as well as antimicrobial and antioxidant activity [22,23,24]. However, most applications of carbon dots are related to their ability to emit and absorb light. The exact mechanism of fluorescence is not entirely understood. It is understood that, depending on the type of CD, it may result from quantum confinement, the presence of numerous surface defects, surface passivation and the presence of certain heteroatoms. The emission spectra of carbon dots can be easily tuned, providing wider opportunities for fluorescence imaging. Absorption behavior will also vary depending on the size of the π-coupled domains, the types of surface groups and the content of oxygen and nitrogen atoms in the cores of the molecules. Typically, however, they show strong absorption in the radiation range from 200 to 400 nm [25,26,27].

### 2.1. Types of Carbon Dots

The classification of CDs is still widely debated. However, based on the structure of the carbon core, the presence of surface functional groups and the resulting differences in the exhibited properties, CDs are increasingly being divided into graphene carbon dots (GQDs), carbon quantum dots (CQDs), carbon nanodots (CNDs) and carbonized polymer dots (CPDs) (Figure 1) [22,28].

Graphene carbon dots (GQDs) are crystalline and anisotropic graphitic structures with cross-sectional dimensions less than 20 nm and heights within 2.5 nm. They consist of one or several (less than 5) graphene layers, connected by chemical groups present at the edges and within the interlayer defect [29,30]. As in multilayer graphene structures, the spacing between sheets is 0.24 nm [25]. GQDs are typically obtained by top-down methods, by cutting larger carbon materials. Their unique optical properties are determined not only by quantum effects due to particle size, but also by the size and number of π-coupled domains and functional groups found on the surface and edges of the layers [30]. Unlike other carbon dots, only sp^2^ hybridization occurs in them. This means that the three bonds in the carbon atoms form a 120° angle and are in one plane, while one electron remains free to form an electron gas [31,32,33].

Carbon quantum dots (CQDs) are spherical and inherently crystalline structures. They can be obtained by both bottom-up and top-down methods, but are more often formed from smaller particles by folding, polymerization and carbonization processes [22,30]. The spacing between the stripes is close to 0.34 nm, which corresponds to the spacing between graphite layers [25,34]. Most of the valence orbitals are in the sp^2^ hybridization state, but a few sp^3^ states are also observed [31].

Carbon nanodroplets (CNDs) are spherical materials that are amorphous in nature [26]. Unlike GQDs and CQDs, they do not have a quantum entrapment effect. Thus, the photoluminescence of these structures is not due to their small particle size, but comes mainly from the numerous defects present on their surface and from the subdomain state in the carbon core. They have a high degree of carbonization and very numerous oxygen groups [22,29]. They exhibit both types of hybridization-sp^2^ and sp^3^, with more sp^3^ forms [31].

Carbonized polymer dots (CPDs) are characterized by a hybrid structure consisting of a carbon core surrounded by polymer chains and functional groups. This structure improves the stability and compatibility of the particles and makes their further modification and functionalization easier to carry out. In the case of CPDs, optical capabilities are mainly dependent on the molecular state and crosslink-enhanced emission (CEE) effect [30]. Like CNDs, they have an amorphous structure, sp^3^ hybridization predominates in them, and no quantum limitation effect is observed [26,31]. Based on the structure of the carbon core, they can be divided into four subclasses (Figure 1). Two of them have an arrangement of carbons similar to CND or CQD, the third consists of small clusters of carbons separated by polymer chains, while the structure of the last subclass is compact and highly dehydrated [29,31].

### 2.2. Obtaining Carbon Quantum Dots (CQDs)

Carbon quantum dots were first obtained in 2004. They were an unexpected byproduct of the synthesis of single-walled carbon nanotubes using the arc discharge method. As a result of this process, intense thermal conditions occurred in the arc region, which led to the decomposition of graphene and the formation of carbon dots. Their presence was noticed during the purification of nanotubes by preparative electrophoresis, where they became visible in the form of stripes emitting blue, green and orange light [35,36,37]. This discovery has generated tremendous interest in the scientific community and has spurred many research teams to explore these structures and to develop more efficient, scalable and cost-effective methods for their production.

An important aspect of research into the synthesis of carbon quantum dots is the control of their properties. By selecting the method and modifying the synthesis parameters, such as reaction time, temperature or component ratio, one can determine their size, shape, purity, stability, cytotoxicity or light emission efficiency, among others [18,19,20,38]. This is crucial for optimizing their applications in various fields of science and technology. Research is also underway to look for alternative raw materials that would be more environmentally friendly. A great potential in the ecological production of carbon dots is seen in bio-waste. They are generated in large quantities primarily in the food, pulp and paper, forestry and agricultural industries, making them an abundant and continuously generated source of carbon [20,38,39]. Studies show that CQDs produced from biomass are less toxic and more biocompatible which is particularly important for their use in the medical and food sectors [40,41,42]. In addition, using them as precursors minimizes the consumption of natural resources and reduces the amount of pollution produced. Products that would have ended up in landfills can be reused to create valuable and highly functional nanomaterials. Also, residues from the CQDs manufacturing process can then serve as a precursor for subsequent applications. The practice of producing nanoparticles from waste thus fits in with the principles of sustainable development and a closed-loop economy. The abundant availability of organic waste makes it much cheaper than other chemicals used in traditional production methods. The processes of synthesizing carbon quantum dots from bio-waste are therefore much more economically attractive [15,43].

The methods of obtaining nanoparticles can be divided into two main categories: top-down and bottom-up methods (Figure 1). For the former, carbon dots are produced by chemical and physical cutting methods for larger carbon materials such as graphite, graphene nanotubes or fullerenes. These include chemical exfoliation of nanomaterials, plasma treatment, laser ablation, electric arc discharge, ultrasonic synthesis, and electrical and chemical oxidation [37,44]. Carbon dots prepared by these methods show excellent photophysical properties and have a more ordered structure and higher purity [45]. However, these approaches are generally much more time-consuming and complex, require rigorous reaction conditions, expensive materials and equipment, and the surface functionalization and passivation steps must be carried out separately [46,47,48]. For these reasons, bottom-up approaches are much more popular. They involve manipulating structures at the atomic or molecular level and gradually constructing larger and larger particles. They are used to produce nanoparticles with defined and homogeneous structures, as they allow more precise control over their size and shape [44,49]. In addition, they allow the use of inexpensive and easily available biomass as feedstock, offer the possibility of a one-step process and require milder reaction conditions. All this makes them much cheaper and easier to adapt on a mass scale [46,50]. The most commonly used bottom-up synthesis techniques include chemical vapor deposition, microwave pyrolysis, hydrothermal and solvothermal reactions [37].

#### 2.2.1. Microwave-Assisted Synthesis

Microwaves (MW) are lying between radio frequencies and infrared radiation, with wavelengths ranging from 0.3–300 GHz [51]. Microwave radiation is commonly used for bottom-up synthesis and functionalization of carbon materials, metal hydroxides and oxides, and other organic and inorganic substances [52]. Its impact differs for various types of materials. Non-polar substances, which include the main components of the atmosphere, such as carbon dioxide, nitrogen and oxygen, are incapable of absorbing microwave radiation, so they do not heat under its influence. Polar molecules absorb MW and convert it into thermal energy by two mechanisms: dipole polarization and ionic conduction [53,54].

The first is based on the rotational motion of dipole particles after microwave irradiation. The molecules try to position themselves in an oscillating electric field. When the field changes, the dipole molecules also change their turn and direction, causing mutual friction and dielectric losses. The macroscopic effect of the reactions taking place is an increase in temperature [51,55]. It is greater the more polar the substance is [53]. However, if the dipole does not have enough time to reposition (too high frequencies) or if the changes in orientation occur too quickly (low-frequency radiation), the temperature increase will not occur.

The second mechanism is based on ionic conductivity. Particles that are in solution in ionic form follow an electric field. As they move, they collide with neighboring particles or atoms, resulting in heat generation [55].

Carbon nanomaterials have a high capacity to absorb microwave radiation and interact with it in many ways. In their case, the presence of a large number of free electrons is responsible for heating. They try to oscillate with a certain phase of radiation, but with continuous and rapid phase changes they cannot keep up with matching it. This phase lag leads to energy dissipation in the form of heat [52]. The syntheses via MW offer many advantages over conventional heating techniques. With thermal methods, heating is accomplished through the processes of convection and conduction of heat from the outside of the sample to the inside. The temperature gradient created can result in incomplete reactions, lower homogeneity and mechanical strength of the materials obtained, and can promote the formation of by-products. Conduction- and convection-based techniques also require longer reaction times. In some cases, prolonged exposure to high temperatures can lead to degradation of the reactants, which can reduce reaction efficiency or completely prevent its occurrence [51,52].

Unlike conventional methods, microwave radiation absorption bypasses the reaction vessels and directly heats the target particles. In this technique, initially the inside of the particle is hotter than its surface. This makes the temperature rise more uniform, allowing processes to occur much faster and with fewer byproducts. This, in turn, reduces the number of steps required to purify the sample. In addition, because radiation can be stopped immediately and lower reaction temperatures can be used, the risk of overheating samples is minimal [36,51]. Another advantage is greater control over the experimental parameters, making it easier to obtain products with the desired and uniform shape, size and morphology [52].

The main limitations of the microwave method are related to the pressure that microwave reactors can reach (~300–400 psi), which also affects the acceptable reaction temperature. Another problem is the volume of reaction samples. Reactors typically operate on small volumes making this method rarely used for mass production [51].

#### 2.2.2. Laser Ablation Technique

Laser techniques have undergone significant development over the years, both in terms of understanding the processes involved in the interaction of a light beam with matter and in improving the performance of lasers. Improvements have been made in the power, precision, stability and temporal accuracy of the devices, among other things. As a result, they are finding increasing applications in many fields, particularly those related to materials and surface engineering [56].

Laser ablation is a method of separating layers of material by treating its surface with high-energy quanta of laser radiation [57]. It is one of the most widely used methods for the top-down production of carbon nanomaterials. It has attracted much interest in the synthesis of carbon quantum dots because of its simplicity and speed. The dots produced by this method have no contaminants and adopt very small sizes (from 2 to 5 nm) [58]. However, the purified nanoparticles do not exhibit fluorescent properties. The particle acquires them only after a process of surface functionalization, e.g., using ethylene glycol [59].

There are several basic mechanisms of laser ablation. The two most commonly described are the photochemical model and the photothermal model. In photochemical ablation, otherwise known as “cold ablation,” photons lower the binding energy of molecules and cause structures to break down. The thermal effect is very limited, as the duration of the laser pulse is shorter than the thermal relaxation time. The treatment ends before the heat generated is transferred to the environment [60]. In the photothermal model, focusing a laser beam on the surface of a substance rapidly raises its temperature, causing the material to directly transition to a gaseous state. The vaporized layer of material then condenses as a thin film on the platform above [61]. To improve the deposition of the layers, it can be heated to a suitable temperature. However, when the goal is to obtain amorphous or nanocrystalline structures, it will be more efficient to cool it down [62].

The main parameter affecting the absorption of the laser beam by the material is the wavelength. The greater it is, the greater are the depth of absorption and the area of focusing, which in turn affects the quantity and size of products generated. The second important factor is the width, understood as the pulse duration. A shorter width ensures faster evaporation and a reduction in the area affected by heat. This results in a more efficient ablation process. The third factor that has a strong influence on the size and shape of the synthesized particles is the state of the laser focus point. The particle size will be largest when the target is set at the focal point, while smaller particle sizes will be obtained when the target is set below the focus [63].

The atmosphere in which the synthesis is carried out is also an important factor influencing the nature of the resulting product. Materials synthesized in vacuum will be characterized by complete purity and high reactivity. However, expensive vacuum equipment is required to achieve such conditions, which makes the use of this technique relatively expensive and problematic. The cheapest atmosphere in which the ablation process can be carried out is air. However, the resulting nanomaterials can be highly toxic. Their purity is highly dependent on the quality of the surrounding air, which can contain various types of sulfides, oxides, nitrides and hydroxides. Controlling the level of aggregation of particles obtained by this technique is also a major problem. Liquids have proven to be the most favorable reaction environment. They are both more cost-effective and accessible than vacuum techniques and can provide relatively good protection against contamination [64].

### 2.3. Applications of Carbon Dots

Research developments leading to increasing control over the synthesis and properties of carbon quantum dots are opening up many new application opportunities for them. The field of their applications is expected to expand further, opening up new prospects for their use in various fields of science and technology. Currently, the greatest potential of carbon dots is seen in sectors related to biotechnology, medicine and the food industry.

#### 2.3.1. Applications of Carbon Dots in Biotechnology

The main applications of carbon quantum dots include bioimaging of cellular structures, determining pH and other parameters of the cellular environment, studying the presence and concentration of small-molecule compounds, and monitoring biochemical processes taking place in cells. All of these analyses are feasible on living and apoptotic cells, both in vitro and in vivo [65,66,67]. The main application directions for CDs in biotechnology are shown in Figure 2.

Selective staining of individual structures found in cells can give insight into their metabolic and proliferative activity, cycle and division phase, and highlight the influence of exogenous factors involved in growth stimulation and regeneration [59]. Moreover, with the help of fluorescence microscopy, it is possible to monitor molecular dynamics in real time. This makes it possible to elucidate many hitherto poorly understood phenomena in cell biology related to translocation and interactions between molecules and responses to external stimuli [25].

Compounds used in bioimaging should meet several basic criteria. First of all, they must exhibit a lack of cytotoxicity, have high selectivity and fluorescence capabilities. Classical organic dyes are easily degraded, are often prone to photobleaching and, in the case of in vitro studies, can cause contamination of the culture medium. This significantly deteriorates the efficiency of the imaging and detection process [68].

Carbon quantum dots, appear to be an excellent solution. They enable the detection of structures and substances with a selectivity unavailable to any other organic molecules, while showing stability, lack of cytotoxic effect and much better biocompatibility compared to metallic quantum dots [59,69]. It has been shown that the factors that have the greatest influence on the ability of particles to enter cells and accumulate within specific structures are the presence of particular functional groups, electrical charge and degree of hydrophobicity. It therefore follows that by properly selecting the synthesis parameters, it is possible to control which cellular elements will undergo staining. For example, carbon dots obtained by the microwave method will accumulate mainly in the cell membrane and close to the cell nucleus [59,65,70]. There is also a lot of research currently underway on the use of carbon quantum dots in monitoring environmental pH. Changes in pH can significantly affect the color and intensity of their fluorescence. Due to their easy handling, short response time, high sensitivity, and applicability to biological environments, they are of particular interest in areas related to medical and biological sciences [25]. Although the mechanism of sensitivity of carbon dots to pH is not fully understood, there are several concepts explaining this relationship. The most widely accepted hypothesis is that base and excited states involve deprotonation or protonation of surface functional groups containing oxygen in their composition. This causes transitions in the energy levels of the particles, which become apparent in changes in their fluorescence. Most carbon dots have pH-sensitive luminescence properties, but depending on the raw materials, method of preparation, structure and morphology, they will respond to changes in different ways [71,72]. Potential examples of the use of carbon dots in biotechnology are presented in Table 1.

#### 2.3.2. Application of Carbon Dots in Food Technology

Food production requires continuous monitoring of the quality of raw materials and the foodstuffs produced from them. At each stage of processing, there is a potential risk of contaminants that can adversely affect product safety. Contaminants can have various sources and include pathogenic microorganisms present in the environment, microbial toxins, heavy metals, biogenic amines, products of the glycosylation process and residues of pesticides and veterinary drugs. The last are a very big problem for meat and grain products [106]. Consumption of such contaminated goods may be a serious risk to human health and life. Heavy metals exhibit particularly toxic effects. It is estimated that about 90% of these elements are introduced into the body with food [28,107]. Some metallic elements, such as lead, chromium and mercury, have devastating effects on health even at low concentrations. Others, which include copper and silver, are not biodegradable and accumulate in internal organs, causing increasingly serious health consequences over time [106].

Traditional analytical methods, which include liquid and gas chromatography and mass spectroscopy, are time-consuming, complex, require expensive equipment and skilled personnel. They are therefore unable to meet the needs of effective real-time monitoring of the entire production chain. Therefore, to ensure food safety, there is an urgent need to develop sensitive, accurate, safe and rapid analytical methods. Detection methods based on fluorescence show great potential here. Of all fluorescent materials, carbon dots are the most promising. They provide satisfactory optical properties and high sensitivity, and unlike other quantum dots are characterized by biocompatibility and lack of cytotoxicity. Their low cost of obtaining is also a great advantage, which facilitates their use on a mass scale [106,108,109].

Food safety at every stage of its production is a key aspect of protecting the health and lives of consumers. Physical, chemical as well as biological contamination of food may not only result from direct contamination of food products, but may also pass into it indirectly. Therefore, it is essential to monitor and take care of the quality of the food production at every stage, in line with the ‘farm-to-table’ strategy. This demands an effective method to guarantee food safety [110,111]. The low toxicity and high sensitivity of CDs makes them suitable for large-scale application. In the food industry, they are applied for the detection and determination of heavy metals, pathogens and additives [112,113,114,115]. The main application directions for CDs in food technology are shown in Figure 3.

The properties of CDs can be tailored already during the synthesis process, mainly by appropriate functionalization of their surface. Depending on the functional groups present, nanoparticles will show affinity for other types of compounds. Mostly, a given type of carbon dots is particularly selective for one compound and the content of other substances will not affect changes in fluorescence intensity. Other features that can be controlled in the synthesis process are the degradation ability to reduce secondary product contamination by these structures and sensitivity to changes in the concentration of the test compound. Combining nanoparticles with other ligands can also be a good option. Such a treatment can significantly improve their sensitivity to changes in the content of target substances and improve their adsorption capacity [110,116,117,118].

Carbon quantum dots can also be used in food preservation. Studies show that they exhibit strong antimicrobial and antioxidant properties, so they can significantly improve product stability and extend shelf life. This is of greatest importance for processed meats, since microbial infections and oxidation processes are the main factors in the loss of quality of these products. Lipid oxidation deteriorates the texture, color and appearance of meat, and secondary products of this process can lead to protein oxidation. This in turn can result in structural changes to these compounds and the formation of toxic substances. However, the antioxidants currently used can negatively affect health and interfere with the original flavor and aroma of the product. CDs are able to assist in combating free radicals while maintaining the sensory value of food items [119,120].

Carbon dots can also be used as nanofillers in packaging materials. Currently, there is a global search for new plastics that are completely biodegradable and at the same time meet all expectations for their functionality. Most composites composed of biopolymers such as polysaccharides or proteins can exhibit many favorable properties and be completely degradable to carbon dioxide, water and biomass. With these types of materials, however, the problem of poor mechanical and barrier properties often arises. Adding even a small amount of carbon nanoparticles to a polymer film can help overcome these limitations without adversely affecting the food. In addition, such packaging can exhibit properties that classify it as active or smart packaging. Active packaging is capable of reducing the rate of spoilage of food products through increased barrier properties and antimicrobial and antioxidant effects. Smart packaging additionally provides information on biochemical and microbiological changes that have occurred in the food. In the case of films containing carbon quantum dots, the appearance of various compounds providing evidence of product deterioration will induce changes in color and fluorescence intensity [24,121,122]. The possibilities for using carbon dots as a smart and active packaging in food technology are much wider. For example, Da et al. [123] analysed the degree of oxidation in bananas and apples using fluorescent coatings on polypropylene film through the photografting mechanism. The quality was evaluated by assessing the number of black spots and areas on the film. Min et al. [124] applied carbon dots as a functional filler to prepare a functional film based on gelatine. The resulting films exhibited UV barrier properties and strong antioxidant activity. Carbon dot based sensors are most commonly based on a fluorescence quenching mechanism induced by the surface states of the carbon dots. On this basis, metal ions such as Ag^+^ [125], Fe^3+^ [126], Hg^2+^ [127], Cu^2+^ [122], Ni^2+^ [128] be detected. Ezati et al. [129] have devised a method for detecting ammonia and its derivatives using specially prepared carbon dot-containing paper. The immediate change in the paper’s colour from yellow to brown allows for the estimation of the freshness of shrimp and meat products. Other potential examples of the use of carbon dots in food technology are presented in Table 2.

## 3. Conclusions

Carbon dots (CDs) are distinguished among many materials by their broad but unique properties, which include good photostability, tunable photoluminescence, high quantum efficiency, excellent biocompatibility, small size, low toxicity, and above all, plentiful and inexpensive sources of acquisition. Thanks to these properties, CDs have gained great application importance in many fields, including biotechnology, food technology, catalysis, optoelectronic devices and environmental protection. It therefore seems crucial to obtain CDs from natural sources and waste biomass. This novel approach to recycling may in the future enable the development of functional materials from waste biomass, and the development of large-scale quantum dot synthesis will have a significant impact on the sustainability of the global economy.

## Figures and Tables

**Figure 1 ijms-24-14984-f001:**
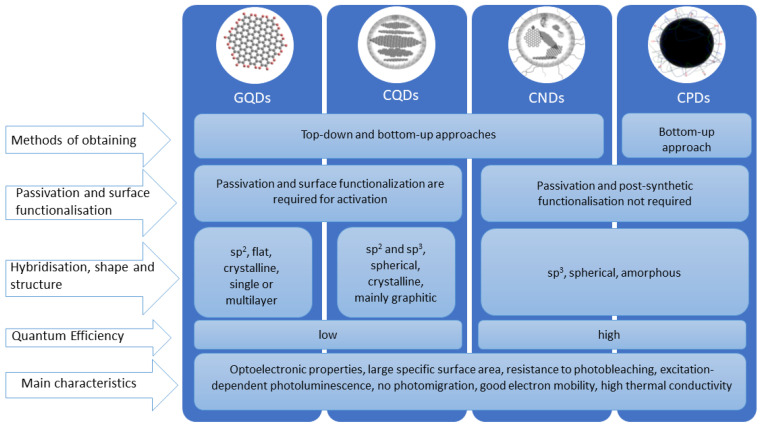
Structure, preparation methods, and basic properties of Carbon Dots.

**Figure 2 ijms-24-14984-f002:**
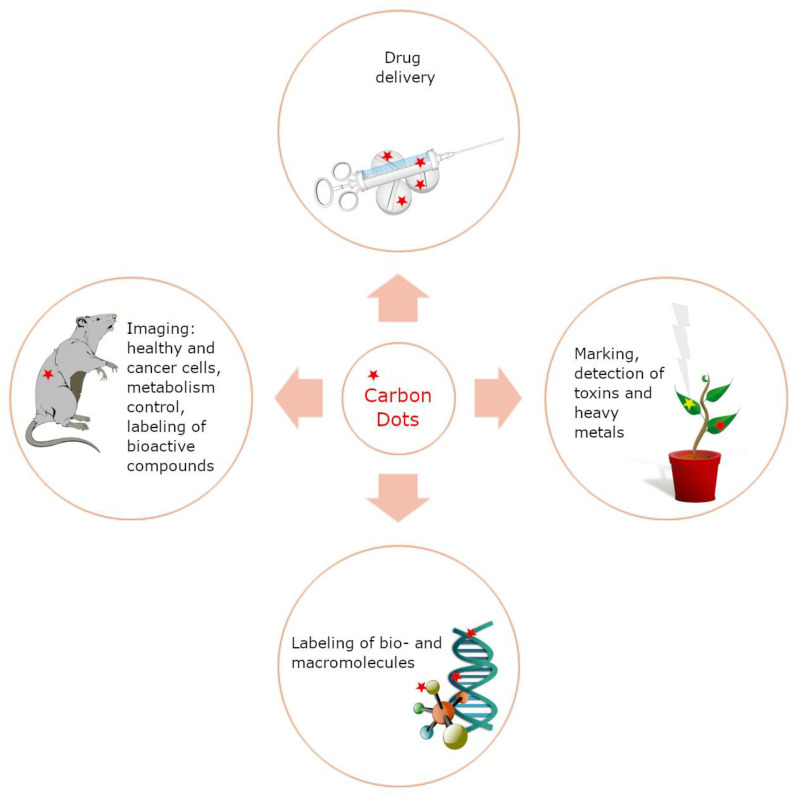
Main directions of application of CDs in biotechnology.

**Figure 3 ijms-24-14984-f003:**
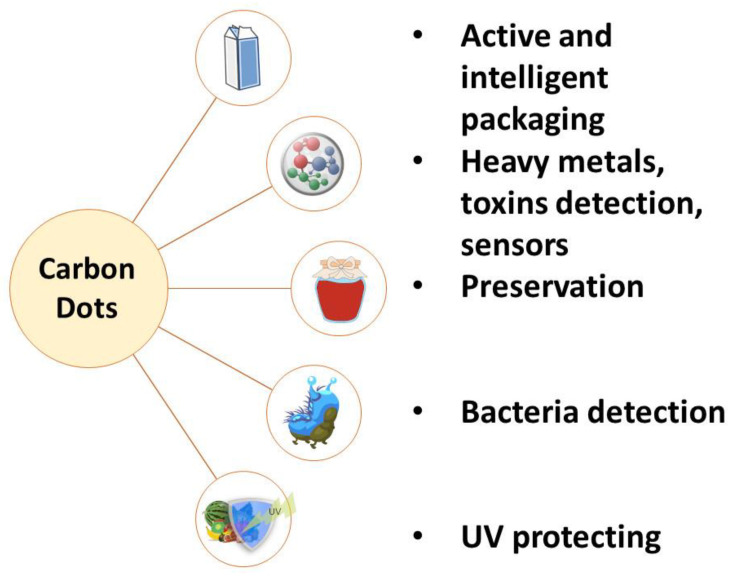
Main directions of application of CDs in food technology.

**Table 1 ijms-24-14984-t001:** Examples of carbon dots application in biotechnology.

Area	Application	References
Drug delivery system	Fluorescent CDs with a carboxyl-rich surface used as a drug delivery system for localized therapy in vivo.	[73]
	Highly fluorescent carbon dots for delivery of doxorubicin and drug delivery system for cancer therapy.	[74]
	Magnetofluorescent carbon dots obtained from waist crab shell for bioimaging and drug delivery.	[75]
	Carbon dots as a model for nanodrug delivery for glioblastoma brain tumors.	[76]
	Carbon dots/hollowmesoporous silica-based platform for near-infrared fluorescent imaging and therapeutic functions.	[77]
	Carbon dots/thermo-sensitive in situ gel as drug delivery agent.	[78]
	Biowaste-derived carbon dots/hydroxyapatite nanocomposite used as drugcarrier for acetaminophen.	[79]
	CDs conjugated with amoxicillin used as a drug delivery agent for bacterial inhibition.	[80]
	Carbon dot-protoporphyrin IX conjugates as carriers in drug delivery and bioimaging applications	[81]
	Nanoplatform of doxorubicin and carbon dots for targeted delivery and controlled discharge of anti-cancer medicines.	[82]
Bioimaging	Carbon dots, synthesized from the root extract of red Korean ginseng used as bioimaging and drug delivery agent to improve antioxidant and antimicrobial activity.	[83]
	Red-emissive carbon dots from aliphatic precursors applied in solid-state micro/nano lasers.	[84]
	CDs used in vivo bio-imaging and in fluorescence bio-labeling of *Escherichia coli* bacteria	[85]
	Malic acid carbon dots used for super-resolution fluorescence localization microscopy.	[86]
	Protein-mimicking CDs for nucleolar stress studies in cell diagnostics and therapeutics.	[87]
Biosensing	CDs as an electrochemical immunosensor for the selective and sensitive determination of the biomarker receptor tyrosine kinase (AXL) in human serum.	[88]
	Biosensor for the assessment of contaminant toxicity in the aquatic environment.	[89]
	Fluorescence detector for the specific detection of Fe^3^+ ions in aqueous solution.	[90]
	CDs-based two-photon fluorescent sensor for monitoring pH variation in living cells and tissues.	[91]
	A nanozyme with high catalytic activity derived from cobalt-doped carbon dots for biosensor and anti-tumour activity.	[92]
	Carbon dot conjugated antibody as a biosensor for progesterone hormone screening.	[93]
	Fluorescent carbon nanodroplets functionalised with ionic liquid for use in electrocatalysis, biosensors and cell imaging.	[94]
	Nitrogen-doped CDs and gold nanoparticles for the detection of cysteine in human serum.	[95]
	Nitrogen doped fluorescent carbon dots to the detection of Hg^2+^ in real water samples.	[96]
	Carbon dots prepared from rice husk to detecting alcohol vapour and of distinguishing between methanol, ethanol, and certain volatile organic compounds	[97]
	CDs from Mahogany fruit shell by chemical oxidation method for the selective detection of d-Penicillamine (D-PA).	[98]
	Carbon dots derived from tea residues, doped with nitrogen, for the detection of tetracycline in urine and pharmaceutical samples and for yeast cell imaging.	[99]
	Carbon dots doped with nitrogen and phosphorus for the identification of nitro explosives.	[100]
	Graphene quantum dots/silicon/ruthenium ion as a fluorescent agent for the detection of triclosan.	[101]
	Carbon dots from tapioca use in wastewater detection and treatment, bioimaging and chemical detection.	[102]
	Aptamer-functionalized nitrogen-doped graphene quantum dots (N-GQDs-aptamer) coupled with cobalt oxyhydroxide (CoOOH) nanoflakes for the detection of tetracycline (antibiotic) residues in foodstuffs.	[103]
	Dual-emission carbon dots for detection of antibiotic residues—penicillin in dairy products.	[104]
	Amino-functionalized QDs combine with for detection of aflatoxin B1.	[105]

**Table 2 ijms-24-14984-t002:** Examples of carbon dots application in food technology.

Area	Application	Form of CDs	References
Food safety	colorimetric detection of H_2_O_2_ and glucose	nitrogen-doped graphene quantum dots (N-GQDs)	[130]
	detection of Ag^+^ ions in water	luminescent Carbon Quantum Dot hydrogels (CQDGs)	[125]
	Cr (VI), Cu (II) and Pb (II) detection	carbon dots@graphitic-carbon nitride (CDs@g-C_3_N_4_) nanocomposite	[131]
	Mn^2+^ detection	glutathione carbon dots-agarose hydrogel film	[132]
	Hg^2+^ detection	CD-based nanohybrid sensor	[133]
	Ag^+^ detection	Nitrogen-Doped Carbon Quantum Dots	[134]
	biosensor for melamine detection	amino-functionalized carbon dots	[135]
	Formalin detection in Fish, shrimp cauliflower, apple	nitrogen-doped carbon dots	[136]
	histamine detection in fish	fluorescence biosensor based on CDs and synthetic peptides	[106]
	Pesticide detection	fluorescence emitting CDs synthesized from cauliflower juice	[137]
	*E. coli* detection	magnetite-carbon dots by lemon and grape fruit extracts	[138]
		amikacin modified fluorescent carbon dots	[139]
	Listeria monocytogenes detection	nitrogen—carbon dots	[140]
	E. coli and Staphylococcus aureus	boronate-based fluorescent carbon dot	[141]
	bioimaging of bacterial and fungal cells (*E. coli*, *Aspergillus aculeatus* and *Fomitopsis* sp.)	carbon dots from *Manilkara zapota* fruits	[142]
	ochratoxin detection	iron-doped porous carbon (MPC) and aptamer-functionalized nitrogen-doped graphene quantum dots (NGQDs-Apt)	[143]
	organophosphorus pesticides in tap water and food detection	nitrogen-doped carbon dots (N-CDs)	[144]
	fluoroquinolones (FQs) and histidine (His) detection in milk	bright yellow fluorescent CDs	[145]
	thiabendazole (TBZ) detection in food	carbon dots (CDs) from Rosemary leaves modified with molecularly imprinted polymers	[146]
	acetylcholinesterase (AChE) activity and organophosphate pesticides (OPs) detection	Nitrogen and chlorine dually-doped carbon dots	[147]
	detection of nitrite (NO^2−^)	CDs synthesize from m-phenylenediamine	[148]
	shrimp freshness monitoring	pH-sensitive green tea-derived carbon quantum dots	[149]
Food control	ascorbic acid in common fruits determination	N, S co-doped carbon dots	[150]
	curcumin determination in food	nitrogen and chlorine dual-doped carbon nanodots (*N*,*Cl*-CDs)	[151]
	CrO_4_^2−^, Fe^3+^, ascorbic acid and L-Cysteine detection in food samples	carbon dots synthesis of waste tea extract	[152]
	folic acid detection	carbon dots derived from lactos	[153]
	Vitamin B_12_ detection	hermally-reduced carbon dot	[154]
	Thiamine detection	Monodispersed CDs synthesize from coconut water	[155]
	monitoring of food freshness	dual-emission carbon quantum dots	[156]
Food storage	active food packaging	chitosan-based carbon quantum dots	[157]
	active and smart food packaging	carbon dots synthesized from microorganisms and food by-products	[121]
	active food packaging	carbon dots with chitosan coating	[158]
	antimicrobial/ultraviolet protective film	nanocellulose film with carbon dots synthesized from lactic acid bacteria	[159]
	UV-blocking of transparent nanocellulose films	CDs-ONC (oxidized nanocellulose) composites	[160]

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
