# Peer review of "Carbon Dots—Types, Obtaining and Application in Biotechnology and Food Technology"

_ijms, 2023, doi:10.3390/ijms241914984_

Round 1
Reviewer 1 Report
In this review manuscript, the authors described the types of CDs, the most important methods to obtain them, and the application of the CDs in biotechnology and food technology. However, many topics are not described in detail. These concerns should be addressed before the publications.
1. For the examples of CDs application in biotechnology, the authors only presented a table to simply describe the applications. If the authors can describe the application the methods/principles with pictures/figures, it will be better for reader to understand their applications.
2. For the examples of CDs application in food technology, the authors also only presented a table to simply describe the applications. I think if CDs are used for sensors, the authors should describe the detection in detail, for instance, the detection method, turn on or turn off sensors, and LOD, etc.
3. Figs 2 and 3 are missing.
4. When “carbon dots” and other words appear in the main text at the first time, give the abbreviations, and then use the abbreviated ones.
The language is easy to read.
Author Response
Authors: We thank the Reviewer for valuable comments on the manuscript. All changes have been applied to the text using the change tracking function. We hope that the revision made has improved the manuscript and that its quality has improved significantly.
Reviewer: In this review manuscript, the authors described the types of CDs, the most important methods to obtain them, and the application of the CDs in biotechnology and food technology. However, many topics are not described in detail. These concerns should be addressed before the publications.
- For the examples of CDs application in biotechnology, the authors only presented a table to simply describe the applications. If the authors can describe the application the methods/principles with pictures/figures, it will be better for reader to understand their applications.
The figures were added to the text according to Reviewer suggestions.
- For the examples of CDs application in food technology, the authors also only presented a table to simply describe the applications. I think if CDs are used for sensors, the authors should describe the detection in detail, for instance, the detection method, turn on or turn off sensors, and LOD, etc.
The manuscript was improved according to Reviewer suggestions.
- Figs 2 and 3 are missing.
It was corrected.
- When “carbon dots” and other words appear in the main text at the first time, give the abbreviations, and then use the abbreviated ones.
It was corrected.

Reviewer 2 Report
This review is potentially interesting and the initial part providing an overview on carbon dots is well done.
However, the abstract and intro preannounce the existence of a second part on specific applications of these nanomaterials in food and biotech, which is developed very poorly in reality. I suggest to amplify this second part, which should be the most relevant in the overall manuscript. The authors should add figures and discuss examples much more than with very basic tables.
After these adjustments and integrations, the review might be ready for publication in IJMS
The abstract needs to be revised for the english spelling.
Author Response
Authors: We thank the Reviewer for valuable comments on the manuscript. All changes have been applied to the text using the change tracking function. We hope that the revision made has improved the manuscript and that its quality has improved significantly.
Reviewers: This review is potentially interesting and the initial part providing an overview on carbon dots is well done.
However, the abstract and intro preannounce the existence of a second part on specific applications of these nanomaterials in food and biotech, which is developed very poorly in reality. I suggest to amplify this second part, which should be the most relevant in the overall manuscript. The authors should add figures and discuss examples much more than with very basic tables.
The figures were added to the text according to Reviewer suggestions and the manuscript was improved.
After these adjustments and integrations, the review might be ready for publication in IJMS
The abstract needs to be revised for the english spelling.
It was corrected.

Reviewer 3 Report
Drawback of the review-article is a lack of representativeness toward areas of potential application of carbon dots (CDs) to study foods. A revision of the text, in the latter context, should be made by the authors.
A review-article devoted to CDs and their application to food safety could be reviewed by the authors (see, ref. [x1].) The same is true for review-article [x2].
[x1] Shi X., Wei W., Fu Z., Gao W., Zhang C., Zhao Q., Deng F., Lu X. Review on carbon dots in food safety applications; (2019), 194, pp. 809 - 821; DOI: 10.1016/j.talanta.2018.11.005
[x2] Luo X., Han Y., Chen X., Tang W., Yue T., Li Z. Carbon dots derived fluorescent nanosensors as versatile tools for food quality and safety assessment: A review; (2020), 95, pp. 149 - 161; DOI: 10.1016/j.tifs.2019.11.017
As potential area of application of CDs to food technology (Table 1) there should be mentioned application to 'food safety', 'food storage', and 'food control'; thus, discussing works [x3]-[x10].
[x3] Lu Z., Su T., Feng Y., Jiang S., Zhou C., Hong P., Sun S., Li C. Potential application of nitrogen-doped carbon quantum dots synthesized by a solvothermal method for detecting silver ions in food packaging
(2019), 16 (14), art. no. 2518; DOI: 10.3390/ijerph16142518
[x4] Yanqi L., Yu W., Liang F. Functionalization of carbon dots and their applications in food safety; (2020), 38 (7), pp. 732 - 740; DOI: 10.3724/SP.J.1123.2019.12003
[x5] Tafreshi F.A., Fatahi Z., Ghasemi S.F., Taherian A., Esfandiari N. Ultrasensitive fluorescent detection of pesticides in real sample by using green carbon dots; (2020), 15 (3), art. no. e0230646; DOI: 10.1371/journal.pone.0230646
[x6] Zhan Y., Zeng Y., Li L., Luo F., Qiu B., Lin Z., Guo L. Ratiometric Fluorescent Hydrogel Test Kit for On-Spot Visual Detection of Nitrite
(2019), 4 (5), pp. 1252 - 1260; DOI: 10.1021/acssensors.9b00125
[x7] Fan H., Zhang M., Bhandari B., Yang C.-H. Food waste as a carbon source in carbon quantum dots technology and their applications in food safety detection; (2020), 95, pp. 86 - 96; DOI: 10.1016/j.tifs.2019.11.008
[x8] Zheng L., Qi P., Zhang D. Identification of bacteria by a fluorescence sensor array based on three kinds of receptors functionalized carbon dots
(2019), 286, pp. 206 - 213; DOI: 10.1016/j.snb.2019.01.147
[x9] Fan K., Zhang M., Fan D., Jiang F. Effect of carbon dots with chitosan coating on microorganisms and storage quality of modified-atmosphere-packaged fresh-cut cucumber; (2019), 99 (13), pp. 6032 - 6041; DOI: 10.1002/jsfa.9879
[x10] Song W., Zhai X., Shi J., Zou X., Xue Y., Sun Y., Sun W., Zhang J., Huang X., Li Z., Shen T., Li Y., Zhou C., Holmes M., Gong Y., Povey M. A ratiometric fluorescence amine sensor based on carbon quantum dot-loaded electrospun polyvinylidene fluoride film for visual monitoring of food freshness; (2024), 434, art. no. 137423; DOI: 10.1016/j.foodchem.2023.137423
A review-article, devoted to application of CDs to latter fiels could be commented on, by the authors, as well (ref. [x11].)
[x11] Zhao J., Liang G., Li A., Man Y., Jin X., Pan L. Review on sensing detection progress of "lean meat agent" based on functional nanomaterials; (2019), 35 (18), pp. 255 - 266; DOI: 10.11975/j.issn.1002-6819.2019.18.031
In Table 2, there should be added application to iron-doped porous carbon (ref. [x12],) in addition to work [x13].
[x12] Wang C., Tan R., Li J., Zhang Z. Exonuclease I-assisted fluorescent method for ochratoxin A detection using iron-doped porous carbon, nitrogen-doped graphene quantum dots, and double magnetic separation
(2019), 411 (11), pp. 2405 - 2414; DOI: 10.1007/s00216-019-01684-7
[x13] Hu Q., Gao L., Rao S.-Q., Yang Z.-Q., Li T., Gong X. Nitrogen and chlorine dual-doped carbon nanodots for determination of curcumin in food matrix via inner filter effect
(2019), 280, pp. 195 - 202; DOI: 10.1016/j.foodchem.2018.12.050
Since, to foods there could be added beverages, then application of CDs to quality control of beverages, including beer, which is described as food, could be added to Table 1 (refs. [x14][x15].)
[x14] Han A., Hao S., Yang Y., Li X., Luo X., Fang G., Liu J., Wang S. Perspective on recent developments of nanomaterial based fluorescent sensors: Applications in safety and quality control of food and beverages; (2020), 28 (4), art. no. 2, pp. 486 - 507; DOI: 10.38212/2224-6614.1270
[x15] Wang H., Liu S., Song Y., Zhu B.-W., Tan M. Universal existence of fluorescent carbon dots in beer and assessment of their potential toxicity, (2019), 13 (2), pp. 160 - 173; DOI: 10.1080/17435390.2018.1530394
The same can be said for application of CDs to drinks, for instance juices. The following work could be cited by the authors [x16].
[x16] Kazemifard N., Ensafi A.A., Rezaei B. Green synthesized carbon dots embedded in silica molecularly imprinted polymers, characterization and application as a rapid and selective fluorimetric sensor for determination of thiabendazole in juices; (2020), 310, art. no. 125812; DOI: 10.1016/j.foodchem.2019.125812
Particularly, there should be mentioned application of CDs to determine pathogenic bacteria, due to their importance for fields of food technology, food control and public health, respectively. For instance, there could be discussed work [x17].
[x17] Sharifi S., Vahed S.Z., Ahmadian E., Dizaj S.M., Eftekhari A., Khalilov R., Ahmadi M., Hamidi-Asl E., Labib M. Detection of pathogenic bacteria via nanomaterials-modified aptasensors; (2020), 150, art. no. 111933; DOI: 10.1016/j.bios.2019.111933
Recent effort on sketching application of CDs to food analysis could be mentioned, as well (see ref. [x18].)
[x18] Yue X.-Y., Zhou Z.-J., Wu Y.-M., Li Y., Li J.-C., Bai Y.-H., Wang J.-L. Application Progress of Fluorescent Carbon Quantum Dots in Food Analysis
(2020), 48 (10), pp. 1288 - 1296; DOI: 10.1016/S1872-2040(20)60049-4
A particularly important area of application of CDs refers to treat food wastes. It is not highlighted by the authors. Given that, work [x19] should be mentioned, accordingly.
[x19] Kechagias A., Lykos C., Karabagias V.K., Georgopoulos S., Sakavitsi V., Leontiou A., Salmas C.E., Giannakas A.E., Konstantinou I. Development and Characterization of N/S-Carbon Quantum Dots by Valorizing Greek Crayfish Food Waste; (2023), 13 (15), art. no. 8730; DOI: 10.3390/app13158730
In addition, there are needed following minor corrections:
1. Row 436: '&ndash' should be deleted. Instead of, there is short dash;
2. Row 467: The first letters of the Journal should be written as capital letters.
3. Row 469: 'j' should be 'J';
4. Row 469: Only the first letter of teh Authors' names should be capital letter;
5. Row 487: '3+' should be superscript;
6. Row 648: '2' of H2O2 should be subscript.
There is not needed an extended editing of the English. However, there are needed technical corrections, particularly, highlighting presentation of the reference section of the manuscript.
Author Response
Reviewer: Drawback of the review-article is a lack of representativeness toward areas of potential application of carbon dots (CDs) to study foods. A revision of the text, in the latter context, should be made by the authors.
Authors: We thank the Reviewer for valuable comments on the manuscript. All changes have been applied to the text using the change tracking function. We hope that the revision made has improved the manuscript and that its quality has improved significantly.
A review-article devoted to CDs and their application to food safety could be reviewed by the authors (see, ref. [x1].) The same is true for review-article [x2].
[x1] Shi X., Wei W., Fu Z., Gao W., Zhang C., Zhao Q., Deng F., Lu X. Review on carbon dots in food safety applications; (2019), 194, pp. 809 - 821; DOI: 10.1016/j.talanta.2018.11.005
[x2] Luo X., Han Y., Chen X., Tang W., Yue T., Li Z. Carbon dots derived fluorescent nanosensors as versatile tools for food quality and safety assessment: A review; (2020), 95, pp. 149 - 161; DOI: 10.1016/j.tifs.2019.11.017
The text of manuscript was improved according to Reviewer suggestions.
As potential area of application of CDs to food technology (Table 1) there should be mentioned application to 'food safety', 'food storage', and 'food control'; thus, discussing works [x3]-[x10].
[x3] Lu Z., Su T., Feng Y., Jiang S., Zhou C., Hong P., Sun S., Li C. Potential application of nitrogen-doped carbon quantum dots synthesized by a solvothermal method for detecting silver ions in food packaging
(2019), 16 (14), art. no. 2518; DOI: 10.3390/ijerph16142518
[x4] Yanqi L., Yu W., Liang F. Functionalization of carbon dots and their applications in food safety; (2020), 38 (7), pp. 732 - 740; DOI: 10.3724/SP.J.1123.2019.12003
[x5] Tafreshi F.A., Fatahi Z., Ghasemi S.F., Taherian A., Esfandiari N. Ultrasensitive fluorescent detection of pesticides in real sample by using green carbon dots; (2020), 15 (3), art. no. e0230646; DOI: 10.1371/journal.pone.0230646
[x6] Zhan Y., Zeng Y., Li L., Luo F., Qiu B., Lin Z., Guo L. Ratiometric Fluorescent Hydrogel Test Kit for On-Spot Visual Detection of Nitrite
(2019), 4 (5), pp. 1252 - 1260; DOI: 10.1021/acssensors.9b00125
[x7] Fan H., Zhang M., Bhandari B., Yang C.-H. Food waste as a carbon source in carbon quantum dots technology and their applications in food safety detection; (2020), 95, pp. 86 - 96; DOI: 10.1016/j.tifs.2019.11.008
[x8] Zheng L., Qi P., Zhang D. Identification of bacteria by a fluorescence sensor array based on three kinds of receptors functionalized carbon dots
(2019), 286, pp. 206 - 213; DOI: 10.1016/j.snb.2019.01.147
[x9] Fan K., Zhang M., Fan D., Jiang F. Effect of carbon dots with chitosan coating on microorganisms and storage quality of modified-atmosphere-packaged fresh-cut cucumber; (2019), 99 (13), pp. 6032 - 6041; DOI: 10.1002/jsfa.9879
[x10] Song W., Zhai X., Shi J., Zou X., Xue Y., Sun Y., Sun W., Zhang J., Huang X., Li Z., Shen T., Li Y., Zhou C., Holmes M., Gong Y., Povey M. A ratiometric fluorescence amine sensor based on carbon quantum dot-loaded electrospun polyvinylidene fluoride film for visual monitoring of food freshness; (2024), 434, art. no. 137423; DOI: 10.1016/j.foodchem.2023.137423
The text of manuscript was improved according to Reviewer suggestions.
A review-article, devoted to application of CDs to latter fiels could be commented on, by the authors, as well (ref. [x11].)
[x11] Zhao J., Liang G., Li A., Man Y., Jin X., Pan L. Review on sensing detection progress of "lean meat agent" based on functional nanomaterials; (2019), 35 (18), pp. 255 - 266; DOI: 10.11975/j.issn.1002-6819.2019.18.031
We have not included two publications for reasons beyond our control. They were not available to us.
In Table 2, there should be added application to iron-doped porous carbon (ref. [x12],) in addition to work [x13].
[x12] Wang C., Tan R., Li J., Zhang Z. Exonuclease I-assisted fluorescent method for ochratoxin A detection using iron-doped porous carbon, nitrogen-doped graphene quantum dots, and double magnetic separation
(2019), 411 (11), pp. 2405 - 2414; DOI: 10.1007/s00216-019-01684-7
[x13] Hu Q., Gao L., Rao S.-Q., Yang Z.-Q., Li T., Gong X. Nitrogen and chlorine dual-doped carbon nanodots for determination of curcumin in food matrix via inner filter effect
(2019), 280, pp. 195 - 202; DOI: 10.1016/j.foodchem.2018.12.050
The table 2 was improved according to Reviewer suggestions.
Since, to foods there could be added beverages, then application of CDs to quality control of beverages, including beer, which is described as food, could be added to Table 1 (refs. [x14][x15].)
[x14] Han A., Hao S., Yang Y., Li X., Luo X., Fang G., Liu J., Wang S. Perspective on recent developments of nanomaterial based fluorescent sensors: Applications in safety and quality control of food and beverages; (2020), 28 (4), art. no. 2, pp. 486 - 507; DOI: 10.38212/2224-6614.1270
[x15] Wang H., Liu S., Song Y., Zhu B.-W., Tan M. Universal existence of fluorescent carbon dots in beer and assessment of their potential toxicity, (2019), 13 (2), pp. 160 - 173; DOI: 10.1080/17435390.2018.1530394
The text, table 1 and 2 have been improved by the proposed references.
The same can be said for application of CDs to drinks, for instance juices. The following work could be cited by the authors [x16].
[x16] Kazemifard N., Ensafi A.A., Rezaei B. Green synthesized carbon dots embedded in silica molecularly imprinted polymers, characterization and application as a rapid and selective fluorimetric sensor for determination of thiabendazole in juices; (2020), 310, art. no. 125812; DOI: 10.1016/j.foodchem.2019.125812
The table 2 has been improved by the proposed references
Particularly, there should be mentioned application of CDs to determine pathogenic bacteria, due to their importance for fields of food technology, food control and public health, respectively. For instance, there could be discussed work [x17].
[x17] Sharifi S., Vahed S.Z., Ahmadian E., Dizaj S.M., Eftekhari A., Khalilov R., Ahmadi M., Hamidi-Asl E., Labib M. Detection of pathogenic bacteria via nanomaterials-modified aptasensors; (2020), 150, art. no. 111933; DOI: 10.1016/j.bios.2019.111933
The text has been improved by the proposed references.
Recent effort on sketching application of CDs to food analysis could be mentioned, as well (see ref. [x18].)
[x18] Yue X.-Y., Zhou Z.-J., Wu Y.-M., Li Y., Li J.-C., Bai Y.-H., Wang J.-L. Application Progress of Fluorescent Carbon Quantum Dots in Food Analysis
(2020), 48 (10), pp. 1288 - 1296; DOI: 10.1016/S1872-2040(20)60049-4
The text has been improved by the proposed references.
A particularly important area of application of CDs refers to treat food wastes. It is not highlighted by the authors. Given that, work [x19] should be mentioned, accordingly.
[x19] Kechagias A., Lykos C., Karabagias V.K., Georgopoulos S., Sakavitsi V., Leontiou A., Salmas C.E., Giannakas A.E., Konstantinou I. Development and Characterization of N/S-Carbon Quantum Dots by Valorizing Greek Crayfish Food Waste; (2023), 13 (15), art. no. 8730; DOI: 10.3390/app13158730
The citation has been added to the text.
In addition, there are needed following minor corrections:
- Row 436: '&ndash' should be deleted. Instead of, there is short dash;
- Row 467: The first letters of the Journal should be written as capital letters.
- Row 469: 'j' should be 'J';
- Row 469: Only the first letter of teh Authors' names should be capital letter;
- Row 487: '3+' should be superscript;
- Row 648: '2' of H2O2 should be subscript.
All of the above comments apply to references that have been made using the end note programmer. In the program settings we have selected the option to quote according to MDPI guidelines. We have sent an enquiry to the Editor if he is able to help us with this issue.
There is not needed an extended editing of the English. However, there are needed technical corrections, particularly, highlighting presentation of the reference section of the manuscript.
It was corrected.

Round 2
Reviewer 2 Report
Now acceptable
Reviewer 3 Report
The authors have addressed all recommendations and remarks on the text. The current, revised, version of the review-article would be in interest in the Readers of the Journal.
There is needed minor corrections of the English.